# Faster Reinforcement Learning with Expert State Sequences

## Abstract

Imitation learning relies on expert demonstrations. Existing approaches often require that the complete demonstration data, including sequences of actions and states are available. In this paper, we consider a realistic and more difficult scenario where a reinforcement learning agent only has access to the state sequences of an expert, while the expert actions are not available. Inferring the unseen expert actions in a stochastic environment is challenging and usually infeasible when combined with a large state space. We propose a novel policy learning method which only utilizes the expert state sequences without inferring the unseen actions. Specifically, our agent first learns to extract useful sub-goal information from the state sequences of the expert and then utilizes the extracted sub-goal information to factorize the action value estimate over state-action pairs and sub-goals. The extracted sub-goals are also used to synthesize guidance rewards in the policy learning. We evaluate our agent on five Doom tasks. Our empirical results show that the proposed method significantly outperforms the conventional DQN method.

## 1 Introduction

Human expert data are widely used for policy learning in sequential decision-making tasks. A common learning paradigm is imitation learning where a policy is trained via the expert supervision to clone the behaviors of the human experts Pomerleau (1989); Ross et al. (2011); Liu et al. (2017). In general, actions performed by human experts are explicitly required as input to an imitation learning algorithm. An alternative to utilizing the human expert data is to convert the data into rewards to guide the policy learning of a sequential decision-making task either via inverse reinforcement learning or reward shaping Abbeel & Ng (2004); Levine et al. (2011). There has been work on combining human expert data and reinforcement learning. For example, Hester et al. (2017) utilizes human expert data to facilitate the learning of action value functions on the ATARI games.

The aforementioned approaches all require that the actions performed by the human expert are known, so that they can be used as input for learning. However, this information is not always available. For example, the expert may have a different action set or the expert may perform at a different temporal resolution. Even the actions performed by the expert are recorded, a reinforcement learning agent still cannot directly clone the expert behaviors. Therefore, in this paper we are interested in the learning scenario where only the state sequences of the expert are known but the actions performed by the expert are missing. Similar setting is investigated by Borsa et al. (2017). Given the expert state sequences, arguably the agent could build a dynamic model of the environments via maximizing the likelihood of the observed expert state sequences. Then the unknown actions of the expert might be inferred. Direct behavior cloning techniques could be possibly applied to obtain a policy. However, building high-fidelity dynamic models for stochastic environments is itself a challenging and data-consuming problem.

Our agent utilizes the expert state sequences without building a model of the environment. Specifically, two consecutive states in the expert state sequences define a favorable direction in the state space to solve the task. This direction information can be viewed as sub-goals in the learning process. The agent can leverage these sub-goals to facilitate learning in several ways. First, the agent could factorize the action value functions over state-action pairs and sub-goals. Different state-action pairs may be combined with similar sub-goals so the learned estimation could be easily generalized.

The sub-goal space may also have good structures to facilitate learning. Second, we can derive additional rewards to guide the policy learning by following the expert state sequences. This kind of guidance rewards could help the agent to mitigate the problem of delayed task-specifying rewards in learning (credit assignment problem). We evaluate our method on 5 Doom tasks and the empirical results show that the proposed method can learn policies achieving superior performance over a conventional DQN method Mnih et al. (2013) by only leveraging expert state sequences.

## 2    SEQUENTIAL DECISION MAKING WITH EXPERT STATE SEQUENCES

We consider sequential decision-making tasks formulated as Markov decision processes (MDPs). An MDP is a tuple $< \mathcal{S}, \mathcal{A}, P, R, \gamma >$ where $\mathcal{S}$ is the state space and $\mathcal{A}$ is the action space. $P : \mathcal{S} \times \mathcal{A} \times \mathcal{S} \to [0, 1]$ is the state transition function and $P(s, a, s') = \Pr(s'|s, a)$ is the probability that the next state is $s'$ given that current state is $s$ and action $a$ is taken. $R : \mathcal{S} \times \mathcal{A} \to \mathbb{R}$ is the reward function with $R(s, a)$ being the expected immediate reward of taking action $a$ in state $s$. $\gamma$ is a discount factor determining the tradeoff between short-term and long-term rewards. A stochastic policy $\pi : \mathcal{S} \times \mathcal{A} \to [0, 1]$ specifies the action to take in states. A state-action value function is defined as $Q^\pi(s, a) = \mathbb{E}[\sum_{t=1}^\infty \gamma^{t-1} r_t | s_0 = s, a_0 = a, \pi]$. The optimal policy $\pi^*$ has action value function $Q^* = \max_\pi Q^\pi$. Taking actions greedily with respect to $Q^*$ yields the optimal policy $\pi^*$.

In the observational learning setting, the agent also has access to some state sequences demonstrated by an expert, but the actions performed by the expert are unknown. Specifically, we denote the state sequences of the expert as a set of $N$ consecutive state pairs $\mathfrak{D} = \{(\hat{s}_i, \hat{s}_i')\}_{i=1}^N$. If the agent ignores the state sequences of the expert completely, the problem reduces to conventional reinforcement learning tasks. Our research interest in this paper is to design an efficient policy architecture $\pi(.|\mathfrak{D})$ which facilitates the expert state sequences.

## 3    MODEL DETAILS

Given the expert state sequences, arguably the agent could build a dynamic model of the environments via maximizing the likelihood of the observed expert state sequences. Then the unknown actions of the expert might be inferred. Direct behavior cloning techniques could be possibly applied to obtain a policy. However, building high-fidelity dynamic models for stochastic environments is itself a challenging and data-consuming problem. Such approaches have never been applied to complex problems.

Our agent utilizes the expert state sequences without building a model of the environment. Two consecutive states in the expert state sequences define a favorable direction in the state space to solve the task. Such kind of direction information can be viewed as sub-goals in the learning process. The agent can leverage this kind of sub-goals to facilitate learning in several ways. First, the agent could factorize the action value function over state-action pairs and sub-goals. Different state-action pairs may be combined with similar sub-goals so the learned estimation could be easily generalized. The sub-goal space may also have good structures to facilitate learning. Second, we can derive additional rewards to guide the policy learning of the agent to follow the expert state sequences. To the best of our knowledge, this kind of guidance rewards could help the agent to mitigate the problem of delayed task-specifying rewards in learning. Our agent builds on these two ideas.

Our agent is illustrated in Figure 1. At the high level, it consists of two key components, an sub-goal extractor and an action value estimator . The role of the sub-goal extractor is to extract sub-goals from the expert state sequences and generate sub-goal representations as well as guidance rewards to the action value estimator . The action value estimator learns a sub-goal conditioned action value function, trained by value-based reinforcement learning methods with the original task-specifying rewards and the guidance rewards. Next we present the architecture details of the sub-goal extractor and the action value estimator , followed by their training objectives.

**The architecture of the sub-goal extractor.**    Given a state $s$, the sub-goal extractor extracts relevant consecutive state pairs from the expert state sequences $\mathfrak{D}$. The sub-goal extractor firstly converts the state $s$ into a vector of length $H$ by using a state encoder function $\phi : \mathcal{S} \to \mathbb{R}^H$. The state encoder function could be a convolutional neural network for a vision-based control task. The sub-goal extractor also converts the expert state sequences data $\mathfrak{D} = \{(\hat{s}_i, \hat{s}_i')\}_{i=1}^N$ into $N$ key-value

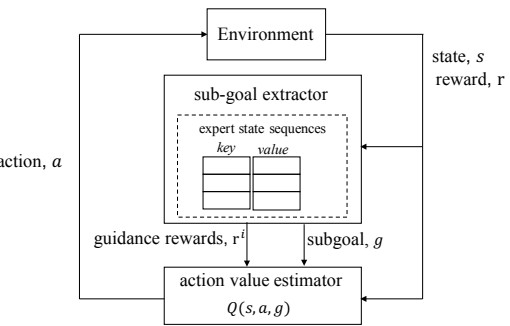

Figure 1: High level view of our agent. Our agent has two key components, a sub-goal extractor and an action value estimator. The sub-goal extractor provides sub-goals and guidance rewards to the action value estimator and the action value estimator learns a sub-goal conditioned action value estimator to interact with the environment.

pairs, where the keys are used to find the match between the agent's current state and the expert states, while the values are sub-goal representation for the consecutive state pairs in the expert state sequences. The $i$-th key is $k_i = W^k \phi(\hat{s}_i)$ where $\hat{s}_i$ is the expert state. The key representation essentially learns the similarity between the agent states and the expert states. The $i$-th value is $v_i = W^v(\phi(\hat{s}_i) - \phi(\hat{s}'_i))$, where $W^k \in \mathbb{R}^{K \times H}$ and $W^v \in \mathbb{R}^{V \times H}$ are learnable parameters and $K$ and $V$ are the key and value vector length. The sub-goal representation for the state $s$ is then a weighted sum of the value vectors followed by a non-linearity ReLU, $\hat{g}(s, \mathcal{D}) = \text{ReLU}(\sum_i \alpha_i v_i)$ and $g = \hat{g}/||\hat{g}||$. The weight $\alpha_i = \text{SoftMax}(W^k \phi(s), k_i)$ is a normalized similarity based on the inner product of the state representation and the keys. In practice we only compute the similarity on the top-K closest keys for the sake of computation efficiency. The top-K closest keys could be efficiently extracted by a KD-tree if the length of key vectors is small. The sub-goal extractor is similar to the key-value memory networks (Miller et al., 2016).

**The architecture of the action value estimator.** The action value estimator is $Q(s, a, g) = gW^a \phi(s) + b^a$, where $W^a \in \mathbb{R}^{V \times H}$ and $b^a \in \mathbb{R}$ are learnable parameters for the action $a$. $g = g(s, \mathcal{D})$ is the output from the sub-goal extractor . We use additional guidance rewards in the learning of the action value estimator to encourage the agent to follow the expert state sequences and to allow different policies learned for different sub-goals. Specifically, our guidance reward is based on the cosine similarity between the agent's trajectories and the state sequences, and at the time step $t$, the immediate guidance reward is $r^i_t = \cos(W^v(\phi(s_{t-1}) - \phi(s_t)), g(s_{t-1}, \mathcal{D}))$.

**Training objective of the sub-goal extractor .** Since the role of the sub-goal extractor is to generate good sub-goal representation for the agent to behave, its training objective is to maximize the expected sum of the discounted task-specifying rewards from the environment, i.e.,

$$\mathcal{U}^{\text{extractor}} = \mathbb{E}\left[\sum_{t=1}^{\infty} \gamma^{t-1} r_t\right]$$

To compute the gradient of $\mathcal{U}^{\text{extractor}}$, we need to know $\Pr(s_{t+1}|s_t, g_t)$ because $\nabla \mathcal{U}^{\text{extractor}} = \mathbb{E}\left[\sum_{t=1}^{\infty} \gamma^{t-1} r_t \sum_{t=1}^{\infty} \nabla \log\left(\Pr(s_{t+1}|s_t, g_t)\right)\right]$. But the exact form of the $\Pr(s_{t+1}|s_t, a_t)$ is unknown so the exact gradient is not available. To approximate the gradients, we use the transition policy gradient in Vezhnevets et al. (2017) to update the parameters of the sub-goal extractor . Transition policy gradient computes the gradients exactly if the transition probability were von Mises-Fisher distribution where $\Pr(s_{t+1}|s_t, g_t) \propto \exp\left(\cos(s_{t+1} - s_t, g_t)\right)$. The transition policy gradient of the sub-goal extractor is then

$$\nabla \mathcal{U}^{\text{extractor}} = \mathbb{E}\left[\sum_{t=1}^{\infty} \nabla \cos\left(W^v(\phi(s_t) - \phi(s_{t+1})), g(s_t, \mathcal{D})\right) R_t\right]$$

where $R_t = \sum_{i=t}^{\infty} \gamma^{i-t} r_i$.

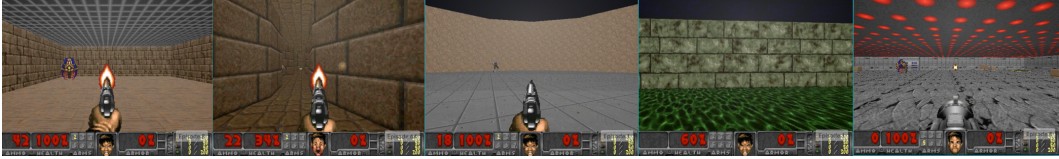

Figure 2: Screen images of Doom tasks. From left to right: DoomBasic, DoomCorridor, DoomDe-fendCenter, DoomHealthGathering and DoomPredictPosition.

**Training objective of the action value estimator .**   We use Deep Q-Network (Mnih et al., 2013) with the truncated $n$-step return to update the parameters of the action value estimator . Specifically, the loss function of the action value estimator is

$$\mathcal{L}^{\text{estimator}} = \mathbb{E}\left[\left(R_t^{(n)} + \gamma^n \max_b Q(s_{t+n}, b, g(s_{t+n}, \mathcal{D})) - Q(s_t, a_t, g(s_t, \mathcal{D}))\right)^2\right]$$

where $R_t^{(n)} = \sum_{k=1}^n \gamma^{k-1}(r_{t+k} + \beta r_{t+k}^i)$ is the truncated $n$-step return and $(s_t, a_t, R_t^{(n)}, s_{t+n})$ are samples from the experience replay buffer in DQN.

## 3.1 REMOVING EXPERT STATE SEQUENCES VIA SELF-TEACHING

The learned policy in our agent depends on the expert state sequences because the sub-goal extractor needs the expert state sequences to generate the input $g(s, \mathcal{D})$ to the action value estimator . In certain control tasks we want the learned policy independent from the expert state sequences. Since our agent has learned a policy which provides the action for each state, an independent policy can be easily derived by direct behavior cloning of the learned policy. Specifically, we collect a dataset of $M$ state-action pairs $\{(s_i, a_i)\}_{i=1}^M$. The states are the last $M$ states in learning the action value function $Q(s, a, g(s, \mathcal{D}))$. The action $a_i$ for a state $s_i$ is the greedy action with respect to the learned action value function, $a_i = \max_a Q(s_i, a, g(s_i, \mathcal{D}))$. This dataset is then used for behavior cloning to derive an independent policy $\mu$. The cross-entropy loss in learning the independent policy is then

$$\mathcal{L}^{\text{self-teach}} = -\sum_i \log\left(\mu(a_i|s_i)\right).$$

## 4 EXPERIMENT SETUP

### 4.1 EMPIRICAL DOMAIN

We evaluate our method on five tasks in the game Doom using OpenAI Gym (Brockman et al., 2016). The screen shots of the tasks are illustrated in Figure 2. In the task of **DoomBasic** the player is spawned in the center of the longer wall and needs to kill a monster spawned randomly on the opposite wall. In the task of **DoomCorridor** the goal of the player is to get to the vest as soon as possible without being killed by the enemies. In the task of **DoomDefendCenter** the goal is to kill as many monsters as possible before running out of ammunition. In the task of **DoomHealthGathering** the goal is to survive by collecting health packs in a map where acidic floor hurts the player periodically. In the task of **DoomPredictPosition** the player needs to use a rocket launcher (with time delay) to kill the monster.

Working directly with raw game screens can be computationally demanding, so we first apply a basic preprocessing step to reduce the input dimension. The raw frames are preprocessed by first converting RGB representation to gray-scale and down-sampling to $80 \times 80$ images. We apply the same preprocessing procedure to all the Doom tasks.

We use a simple frame-skipping technique to play the Doom games. The agent sees and selects actions on every 4-th frame instead of every frame, and its last action is repeated on skipped frames.

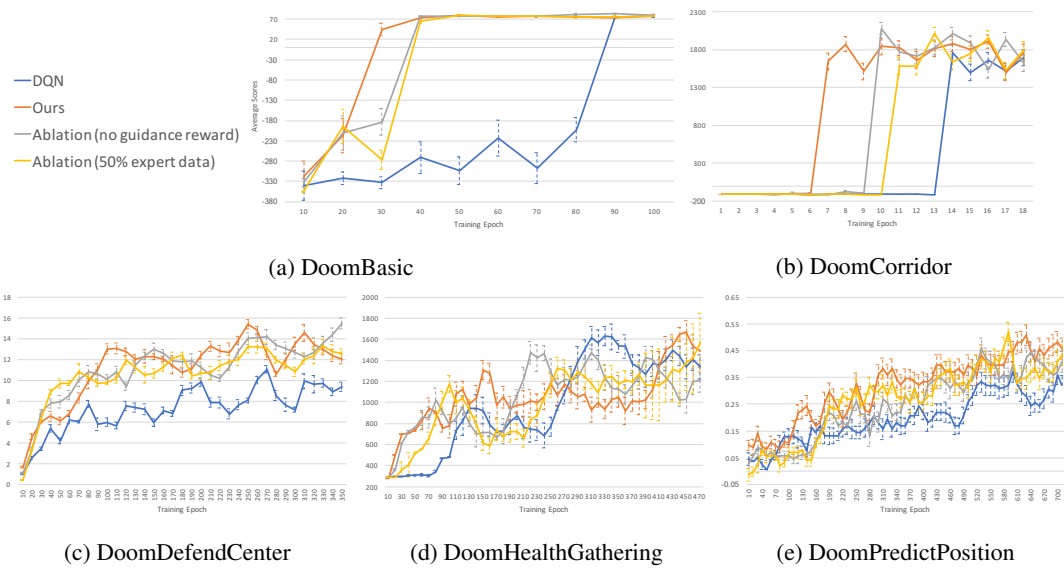

Figure 3: Learning curves of our agents and the DQN baseline. The average game scores are evaluated on the learned model at the end of an epoch with 0.05-greedy policy. The bars are the standard error.

## 4.2 ARCHITECTURE DETAILS

The state encoding function, $\phi$, is a four-layer convolutional neural network. The first hidden layer convolves 32 5×5 filters followed by a 3×3 pooling with stride 2. The second layer convolves 32 3×3 filters followed by a 3×3 pooling with stride 2. The third layer convolves 64 2×2 filters also followed by 3×3 pooling with stride 2. The last layer of the state encoding function is fully-connected and consists of 64 output units ($H = 64$). Each layer is followed by ReLU as nonlinearity.

The learnable matrices, $W^k \in \mathbb{R}^{K \times H}$, $W^v \in \mathbb{R}^{K \times H}$ and $W^a \in \mathbb{R}^{K \times H}$ are set with dimension $K = 16$ and $V = 8$. We use KD-tree to extract top-5 closest keys for sub-goal representation computation.

**DQN baselines.** The DQN baselines have the same state encoder function, $\phi()$, and the outputs are then transformed by a linear layer $Q(s, a) = W^{\text{DQN}}\phi(s) + b$, where $W^{\text{DQN}} \in \mathbb{R}^{|\mathcal{A}| \times H}$ and $b \in \mathbb{R}^{|\mathcal{A}|}$ are learnable parameters and $|\mathcal{A}|$ is the number of available actions.

**LfD baselines.** We also compare to the Learning from Demonstration (LfD) approach. LfD agent would have access to the expert's actions in addition to the state sequences. The LfD agent is only trained on the collected expert data. The architecture of the LfD agent is the same as the DQN baseline except that the final output is normalized by a SoftMax layer. Note that the LfD baselines violate the setting that the expert actions are not observed.

## 4.3 TRAINING DETAILS

We use pre-trained DQN agents with 0.05-greedy policy to collect expert state sequence data. The collection terminates after 5,000 episodes or the number of collected state pairs is 10,000.

We use the truncated 5-step return in learning. The parameters are updated using Adam with default parameters except the learning rate. The learning rates are tuned to have best-performing DQN baselines: DoomBasic(0.001), DoomCorridor(0.0005), DoomDefendCenter(0.001), DoomHealthGathering(0.0005) and DoomPredictPosition(0.0003).

The parameters are updated over 200 minibatches each epoch. Each minibatch consists of 64 samples. We use $\epsilon$-greedy to select actions. The value of $\epsilon$ starts with 1 and it is multiplied by 0.95 every

| Agents | Basic | Corridor | DefendCenter | Health | PredictPosition |
|---|---|---|---|---|---|
| DQN | 82.5 (2.0) | 2071 (78.9) | 11.82 (0.6) | 1628 (104) | 0.37 (0.03) |
| LfD | 82.75 (1.4) | 1940 (93.8) | 10.57 (0.5) | 1241 (119) | 0.32 (0.03) |
| Ours (no Self-Teach) | 83.0 (1.5) | 2093 (70.6) | 15.38 (0.6) | 1644 (95) | 0.52 (0.04) |
| Ours | 83.75 (1.3) | 2015 (70.2) | 15.43 (0.4) | 1615 (102) | 0.50 (0.04) |

Table 1: Performance (game scores) of our agents and the DQN baseline. The average game scores are evaluated on the best-performing learned model in learning. The numbers in parentheses are standard errors.

epoch. The minimum $\epsilon$ value is set to be 0.05. At the end of each epoch, the tasks are evaluated using 0.05-greedy policy with respect to the learned action value estimates. The learning terminates after 700 epochs. The last 10,000 states are collected for the self-teaching procedure to derive independent policy.

## 5 EXPERIMENT RESULTS

First we present the overall performance, followed by ablation studies to help understand the proposed agent. In Figure 3 we compare the learning curves of our agent with the DQN baseline. In four out of the five tasks (Basic, Corridor, DefendCenter and PredictPosition), our method outperforms the DQN significantly: our method learns faster than DQN and converges steadily. Even when considering the expert data in the memory, which corresponds to 1 epoch training samples at top, our method still learns faster. In the task HealthGathering, our method learns faster than the DQN baseline only at the beginning of training, and then performs similarly as the DQN baseline. The agent in HealthGathering needs to navigate in a 3D maze environment and this task has significantly higher degree of partial observability compared to other tasks. Our neural architecture extracts the sub-goals directly from the observation space and the sub-goal space of the task HealthGathering has less useful structure to learn good action value estimates due to the high-degree partial observability.

Table 1 summarizes the performance of our agent after applying self-teaching to remove the dependence on expert state sequences. After self-teaching the performance is similar to the learned policy with expert state sequences in memory. Direct behavior cloning achieves close performance to the expert DQN except on the task HealthGathering.

**Ablation study on number of trajectories.** We apply ablation study to analyze the number of expert trajectories. To this end, we randomly discard fifty percent of the collected consecutive state pairs before the learning starts. The learning curves of this kind of ablative agent is in Figure 3. The impact is task-specific. In the task of Basic and Corridor, the performance degeneration is significant. After discarding expert data, the sub-goal extractor would generate less flexible sub-goals for the action value estimator in the task of Basic and Corridor. In the worst case where there is no expert data, our agent degenerates to a normal DQN architecture with factorized action outputs. This architecture can only generate constant sub-goals and it is even less flexible.

**Ablation study on guidance rewards.** The action value estimator utilizes the sub-goals from the sub-goal extractor in two ways: the first is factorizing action-value estimation with the sub-goals, and the second is the guidance reward to encourage our agent to follow the expert state sequences. In this ablation study we remove the guidance reward in the training loss of the action value estimator $\mathcal{L}^{\text{estimator}}$ and keep everything else unchanged. The performance of this ablative agent is in Figure 3. The performance degenerates on three games significantly (Basic, Corridor and PredictPosition), confirming the positive role of utilizing guidance rewards.

**Illustration of Learned Correlation between Behavior and Expert Trajectories.** It is hard to directly demonstrate that the agent has learned to correlate the learned behavior with the expert state sequences, due to the stochasticity in the environment and the divergence in behaviors. Therefore, we quantitatively investigate the correlation by randomly dropping out expert data in the memory after learning. In learning our agent still has access to all the state sequence data but in testing part of the data are dropped. The rationale is that if our agent has learned to correlate its behavior with the

| Percentage | Basic | Corridor | DefendCenter | Health | PredictPosition |
|---|---|---|---|---|---|
| 100% | 83.0 (1.5) | 2093 (70.6) | 15.38 (0.6) | 1644 (95) | 0.52 (0.04) |
| 50% | -202.6 (30.6) | 1499 (117) | 12.05 (0.3) | 1419 (109) | 0.35 (0.04) |
| 10% | -340.5 (15.4) | -73 (16.2) | 0.85 (0.1) | 740 (70) | 0.04 (0.02) |

Table 2: Performance of the learned policy in different drop rate setting. The rows correspond to different remaining percentage of expert state sequences. The first row is the learned policy without any expert state sequence dropping. The numbers in parentheses are standard errors.

expert state sequences, its performance would degenerate if some of the expert state sequences are removed. The degeneration in the performance is an indicator of the learned correlation. The results are in Table 2. As more data are removed in the memory, the performance degrades accordingly. It implies that our agent is able to extract different subgoals for different states. As some memory state pairs are removed, the agent's performance degrades due to the missing sub-goals.

# 6 RELATED WORK

The problem of learning from demonstrations (also imitation learning) focuses on learning to perform a task by observing an expert, which has attracted considerable research attention from many fields in machine learning. The developed algorithms can largely be divided into two groups of approaches, which are behavioral cloning and inverse reinforcement learning (IRL), respectively. Survey articles include (Schaal, 1999; Billard et al., 2008; Argall et al., 2009).

Behavioral cloning casts the problem as supervised learning, where the learner directly regresses onto the optimal policy of the expert (Pomerleau, 1989; Ross et al., 2011; Liu et al., 2017). However, it assumes the agent receives examples of observation-action tuples, which cannot be applied when action information is absent. On the other hand, inverse reinforcement learning (Ng et al., 2000; Abbeel & Ng, 2004; Ziebart et al., 2008; Levine et al., 2011; Borsa et al., 2017) methods aim to infer the goal of an expert. In other words, it learns a reward function from expert demonstrations, which can then be used in reinforcement learning to recover a policy (Ratliff et al., 2006; Ramachandran & Amir, 2007). More recent methods alternate the step of forward and IRL to obtain more accurate estimates (Ho & Ermon, 2016; Finn et al., 2016). Although the methods in both lines of research have been successfully applied to a variety of tasks, they almost always assume that the provided state-action trajectories of experts are in the same space as the learner observed. As argued in some of the recent work (Stadie et al., 2017; Duan et al., 2017; Liu et al., 2017), such an assumption is restrictive and unrealistic. Instead, they proposed to discover the transformations between the learner and the teacher state space.

Beyond direct imitation learning, there has been some work that combine human expert data and reinforcement learning in policy learning. Gilbert et al. (2015), Lipton et al. (2016) and Lakshminarayanan et al. (2016) store the expert state-action pairs in the experience replay to facilitate learning. Hosu & Rebedea (2016) utilize the state action pairs from a human expert to facilitate the action value learning. Subramanian et al. (2016) leverages human data for efficient exploration. Hester et al. (2017) combines several approaches and demonstrates superior performance on ATARI games. Our method differs from these approaches in that we do not assume human expert actions are available.

# 7 CONCLUSION AND FUTURE WORK

We propose to facilitate the sequential decision-making tasks by utilizing demonstrations from human experts, while we focus on a more realistic scenario where the actions performed by the experts are unavailable. To better make use of the state-only demonstrations, our model learns to extract useful sub-goal information from the expert state sequences to enhance action value estimation. The empirical results on 5 Doom tasks show that our method accelerate the convergence speed. Future work includes extension of the model to partially-observed demonstrations where the observations are not poor approximation of the states.

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
