# OpenReview forum: "Faster Reinforcement Learning with Expert State Sequences"
_ICLR.cc/2018/Conference — Reject_

### Official Review · AnonReviewer1 · 2017-11-25
**A novel approach to learning from state trajectories with some unmotivated details and confusing exposition**

**Rating:** 6
**Confidence:** 3

**Review:**

SIGNIFICANCE AND ORIGINALITY:

The authors propose to accelerate the learning of complex tasks by exploiting traces of experts.
Unlike the most common form of imitation learning or behavioral cloning, the authors
formulate their solution in the case where the expert’s state trajectory is observable,
but the expert’s actions are not. This is an important and useful problem in robotics and other
applications. Within this specific setting the authors differentiate their approach from others
by developing a solution that does NOT estimate an explicit dynamics model ( e.g.,  P( S’ | S, A ) ).
The benefits of not estimating an explicit action model are not really demonstrated in a clear way.

The author’s articulate a specific solution that provides heuristic guidance rewards that cause the
learner to favor actions that achieve subgoals calculated from expert behavior
and refactors the representation of the Q function so that it
has a component that is a function of the subgoal extracted from the expert.
These subgoals are linear functions of the expert’s change in state (or change in state features).
The resultant policy is a function of the expert traces on which it depends.
The authors show they can retrain a new policy that does not require the expert traces.
As far as I am aware, this is a novel approach to the problem.
The authors claim that this factorization is important and useful but the paper doesn’t
really illustrate this well.

They demonstrate the usefulness of the algorithm against a DQN baseline on Doom game problems.
The algorithm learns faster than unassisted DQN as shown by learning curve plots.
They also evaluate the algorithms on the quality of the final policies for their approach, DQN,
and  a supervised learning from demonstration approach ( LfD ) that requires expert actions.
The proposed approach does as well or better than competing approaches.


QUALITY

Ablation studies show that the guidance rewards are important to achieving the improved performance of the proposed method which is important confirmation that the architecture is working in the intended way. However, it would also be useful to do an ablation study of the “factorization” of action values.  Is this important to achieving better results as well or is the guidance reward enough? This seems like a key claim to establish.


CLARITY

The details of the memory based kernel density estimation and neural gradient training seemed
complicated by the way that the process was implemented. Is it possible to communicate
the intuitions behind what is going on?

I was able to work out the intuitions behind the heuristic rewards, but I still don’t clearly get
what the Q-value factorization is providing:

To keep my text readable, I assume we are working in feature space
instead of state space and use different letters for learner and expert:

   Learner: S = \phi(s)
   Expert’s i^th state visit:  Ei = \phi( \hat{s}_i }  where Ei’ is the successor state to Ei

The paper builds upon approximate n-step discrete-action Q-learning
where the Q value for an action is a linear function of the state features:

    Qp(S,a) = Wa S + Ba

where parameters p = ( Wa, Ba ).

After observing an experience ( S,A,R,S’ ) we use Bellman Error as a loss function to optimize Qp for parameter p.
I ignore the complexities of n-step learning and discount factors for clarity.

    Loss = E[    R + MAXa’ Qp(S’,a’)    -   Qp(S,a)   ]

The authors suggest we can augment the environment reward R
with a heuristic reward Rh proportional to the similarity between
the learner “subgoal" and the expert “subgoal" in similar states.

The authors propose to use cosine distance between representations
of what they call the “subgoals” of learner and expert.
A subgoal is defined as a linear transformation of the distance traveled by an agent during a transition.
The heuristic reward is proportional to the cosine distance between the learner and expert “subgoals"

   Rh = B  <   Wv LearnerDirectionInStateS,
                     Wv ExpectedExpertDirectionInStatesSimilarToS   >

The learner’s direction in state S is just (S-S’) in feature space.

The authors model the behavior of the expert as a kernel density type approximator
giving the expected direction of the expert starting from a states similar to the one the learner is in.
Let < Wk S, Wk Ej > be a weighted similarity between learner state features S and expert state features Ej
and Ej’ be the successor state features encountered by the expert.
Then the expected expert direction for learner state S is:

     SUMj  < Wk S, Wk Ej > ( Ej - Ej’ )

Presumably the linear Wk transform helps us pick out the important dimensions of similarity between S and Ej.

Mapping the learner and expert directions into subgoal space using Wv, the heuristic reward is

   Rh = B <   Wv (S-S’),
                    Wv SUMj  < Wk S, Wk Ej > ( Ej - Ej’ ) >

I ignore the ReLU here, but I assume that is operates element-wise and just clips negative values?
There is only one layer here so we don’t have complex non-linear things going on?

In addition to introducing a heuristic reward term, the authors propose to alter the Q-function
to be specific to the subgoal.

   Q( s,a,g ) = g(S) Wa S + Ba

The subgoal is the same as the first part, namely a linear transform of the expected expert direction in
states similar to state S.

    g(S) =  Wv   SUMj  < Wk S, Wk Ej >  ( Ej - Ej’ )

So in some sense, the Q function is really just a function of S, as g is calculated from S.

    Q( S,a ) = g(S) Wa S + Ba

So this allows the Q-function more flexibility to capture each subgoal in a different linear space?
I don’t really get the intuition behind this formulation. It allows the subgoal to adjust the value
of the underlying model? Essentially the expert defines a new Q-value problem at every state
for the learner? In some sense are we are defining a model for the action taken by the expert?


ADDITIONAL THOUGHTS

While the authors compare to an unassisted baseline, they don’t compare to methods that use an action model
which is not a fatal flaw but would have been nice.

One can imagine there might be scenarios where the local guidance rewards of this
form could be problematic, particularly in scenarios where the expert and learner are not identical
and it is possible to return to previous states, such as the grid worlds the authors discuss:
If the expert’s first few transitions were easily approximable,
the learner would get local rewards that cause it to mimic expert behavior.
However, if the next step in the expert’s path was difficult to approximate,
then the reward for imitating the expert would be lower.
Would the learner then just prefer to go back towards those states that it can approximate and endlessly loop?
In this case, perhaps expressing heuristic rewards as potentials as described in Ng’s shaping paper might solve the problem.


PROS AND CONS

Important problem generally. Avoiding the estimation of a dynamics model was stated as a given, but perhaps more could be put into motivating this goal. Hopefully it is possible to streamline the methodology section to communicate the intuitions more easily.

---

> ### Author Response · Authors · 2017-12-22
> **Author Responses**
>
> Thank you for your review. The responses to the questions asked are itemized below:
>
> -- However, it would also be useful to do an ablation study of the “factorization” of action values. Is this important to achieving better results as well or is the guidance reward enough?
>
> We have the ablation results in Figure 3. The ``ablation (no guidance reward)’’ is the agent where only factorization is used and the guidance reward is not used.
>
> -- There is only one layer here so we don’t have complex non-linear things going on?
>
> When we compute the Q values by multiplying the image features (\phi(s)), W^{a} and demonstration features (g), nonlinearity is not used.
>
> -- This allows the Q-function more flexibility to capture each subgoal in a different linear space? I don’t really get the intuition behind this formulation.
>
> By utilizing the sub-goals from the demonstration, we are able to propagate learning experience through successive states in the demonstration trajectories as well as the agents’ experience trajectories.
>
> -- Would the learner then just prefer to go back towards those states that it can approximate and endlessly loop?
>
> The external rewards are also provided to the agent and the endless looping behavior is not optimal with respect to the external rewards.

---

### Official Review · AnonReviewer3 · 2017-11-26
**Works for deterministic controlable dynamics, lack of related work**

**Rating:** 5
**Confidence:** 5

**Review:**

The paper presents a method that leverages demonstrations from experts provided in the shape of sequences of states (actually, state transitions are enough, they don't need to come in sequences) to faster learn reinforcement learning tasks. The authors propose to learn subgoals (actually local rewards) to encourage the agent to go towards the same direction as the expert when encountering similar states. The main claimed advantage is that it doesn't require the knowledge of the actions taken by the expert, only observations of states.

To me, there is a major flaw in the approach. Ho and Ermon 2016 extensively study the fact that imitation is not possible in stochastic environment without the knowledge of the actions. As the author say, learning the actions from state transitions in a standard stochastic MDP would require to learn the model. Yet, the authors demonstrate their approach in environments where the controlable dynamics is mainly deterministic (if one decides to turn right, the agents indeed turns right). So by subtracting features from successive states, the method mainly encodes the action as it almost encodes the one step dynamics in one shot.

Also the main assumption is that there is an easy way to compute similarity between states. This assumption is not met in the HealthGathering environment as several different states may generate very similar vision features. This causes the method not to work. This brings us back to the fact that features encoding the actual dynamics, potentially on many consecutive states (e.g. feature expectations used in IRL or occupancy probability used in Ho and Ermon 2016), are mandatory.

The method is also very close to the simplest IRL method possible which consists in placing positive rewards on every state the expert visited. So I would have liked a comparison to that simple method (using similar regression technique to generalize over states with similar features).

Finally, I also think that using expert data generated by a pre-trained network makes the experimental section very weak. Indeed, it is unlikely that this kind of data can be obtained and training on this type of data is just a kind of distillation of the optimal network making the weights of the network close to the right optimum. With real data, acquired from humans, the training is likely to end up in a very different minima.

Concerning the related work, the authors didn't mention the Universal Value Function Approximation (Schaul et al, @ICML 2015) which precisely extends V and Q functions to generalize over goals. This very much relates to the method used to generalize over subgoals in the paper. Also, the state if the art in IRL and learning from demonstration is lacking a lot of references. For instance, learning via RL + demonstrations was already studied into papers by Farahmand et al (APID, @NIPS 2013), Piot et al (RLED, @ ECML 2014) or Chemali & Lazaric (DPID, @IJCAI 2015) before Hester et al (DQfD @AAAI 2018). Some work is cited in the wrong context. For instance, Borsa et al 2017 doesn't do inverse RL (as said in the related work section) but learn to perform a task only from the extrinsic reward provided by the environment (as said in the introduction). BTW, I would suggest to refer to published papers if they exist instead of their Arxiv version (e.g. Hester et al, DQfD).

---

> ### Author Response · Authors · 2017-12-22
> **Author Responses**
>
> Thank you for your review. The major point you raised was about the technical soundness of our approach in the context of Ho and Ermon 2016. Our problem setting is still in reinforcement learning but not imitation learning. Our results are not trying to support imitation learning is possible given only states.
>
> Regarding the potential issue of using pre-trained network in collecting demonstration data, the network used in collecting the demonstration is the standard DQN and it is different from the architecture of our proposed approach.
> We appreciate the proposed the simple IRL baseline and the pointed papers. We will include them in a revision.
>
> We appreciate the proposed simple IRL baseline and the pointed papers. We will include them in a revision.

---

### Official Review · AnonReviewer2 · 2017-11-27
**This paper proposes a hierarchical approach for speeding up RL through subgoals learned from expert demonstrations. Experiments on the game Doom using OpenAI Gym show some improvements in learning rate over DQN.**

**Rating:** 6
**Confidence:** 4

**Review:**

The authors propose to speed up RL techniques, such as DQN, by utilizing expert demonstrations. The  expert demonstrations are sequences of consecutive states that do not include actions, which is closer to a real setting of imitation learning. The goal of this process is to extract a function that maps any given state to a subgoal. Subgoals are then used to learn different Q-value functions, one per subgoal.
To learn the function that maps states into subgoals, the authors propose a surrogate reward model that corresponds to the angle between: the difference between two consecutive states (which captures velocity or direction) and a given subgoal. A von Mises- Fisher distribution policy is then assumed to be used by the expert to generate actions that guide the agent toward the subgoal. Finally, the mapping function state->subgoal is learned by performing a gradient descent on the expected total cost (based on the surrogate reward function, which also has free parameters that need to be learned).
Finally, the authors use the DQN platform to learn a Q-value function using the learned  surrogate reward function that guides the agent to specific subgoals, depending on the situation.
The paper is overall well-written, and the proposed idea seems interesting. However, there are rather little explanations provided to argue for the different modeling choices made, and the intuition behind them. From my understanding, the idea of subgoal learning boils down to a non-parametric (or kernel) regression where each state is mapped to a subgoal based on its closeness to different states in the expert's demonstration. It is not clear how this method would generalize to new situations. There is also the issue of keeping tracking of a large number of demonstration states in memory. This technique reminds me of some common methods in learning from demonstrations, such as those using GPs or GMMs, but the novelty of this technique is the fact that the subgoal mapping function is learned in an IRL fashion, by tacking into account the sum of surrogate rewards in the expert's demonstration.
The architecture of the action value estimator does not seem novel, it's basically just an extension of DQN with an extra parameter (subgoal g).
The empirical evaluation seems rather mixed. Figure 3 shows that the proposed method learns faster than DQN, but Table I shows that the improvement is not statistically significant, except in two games, DefendCenter and PredictPosition. Are these the results after all agents had converged?
Overall, this is a good paper, but focusing on only a single game (Doom) is a weakness that needs to be addressed because one cannot tell if the choices were tailored to make the method work well for this game. Since the paper does not provide significant theoretical or algorithmic contribution, at least more realistic and diverse experiments should be performed.

---

> ### Author Response · Authors · 2017-12-22
> **Author Responses**
>
> Thank you for your review. Table 1 shows the final performance of the learned models. Our claim of speeding up learning is about the learning curves in Figure 3.

---

### Public Comment · (anonymous) · 2017-11-13
**Open source code**

Hello, I am working on reproducing your work in “Faster Reinforcement Learning with Expert State Sequences” for the ICLR 2018 Reproducibility Challenge. I was wondering if you were planning to open source the code used to perform your experiments, and if you are when it might be available.

Thanks and best regards

---

> ### Author Response · Authors · 2017-11-14
> **Open source code**
>
> Thanks for your interest in our paper. We will release the source codes after the internal approval.

---

### Public Comment · (anonymous) · 2017-12-21
**summary of a reproducibility study on this paper**

Reproducibility of work in machine learning is critical to assuring the veracity of research results.
This is a challenge that has become increasingly important due to the expanding space of model architectures and the evaluation environments used for comparing these models.

We attempted to reproduce experiments in "Faster Reinforcement Learning with Expert State Sequences" (FRLWESS), comparing the learning speed of a DQN baseline against the author's proposed algorithm. The claim of this paper is that using the proposed algorithm for learning from expert state sequences we can learn faster than an agent without expert knowledge and that we can achieve the same performance as state-of-the-art imitation learning models without using expert action information.

However, we were only able to partially reproduce results discussed in the paper. The challenges we faced included difficulty in finding the correct values for certain parameters used in the original paper, such as the size of the experience replay buffer, the constant term $\beta$ for weighting guidance rewards, and the update rules for the expert states dictionary used in the sub-goal extractor.

While we were not able to replicate the results successfully. We would like to share our approach and the source code we used to replicate the experiments, which can be found here: https://www.overleaf.com/read/gntbmmpqwykr.
Our approach to reproducing this paper involved three experiments.
(1.) The first was implementing the DQN agent which was used as a baseline for comparison against FRLWESS.
(2.) The second was implementing and testing the FRLWESS algorithm itself to see if we could replicate the results presented by its authors. As a requirement for running this experiment, we had to collect state sequences from our expert pre-trained DQN model for the imitation learning agent to learn from.
(3.)  Finally, we attempted to replicate the ablation study of the guidance rewards for training the action value estimator in FRLWESS.

---

### Decision · Program_Chairs · 2018-01-29
**ICLR 2018 Conference Acceptance Decision**

**Decision:**

Reject

**Comment:**

This paper proposes a simple idea for using expert data to improve a deep RL agent's performance. Its main flaw is the lack of justification for the specific techniques used. The empirical evaluation is also fairly limited.